# The Meaning of Successful School Leadership in Disadvantaged Contexts in Spain: Approach from the International Successful School Principalship Project (ISSPP)

Cristina Moral-Santaella *[ID] and Francisco Raso-Sánchez

Department of Didactics and School Organization, Faculty of Education, University of Granada, 18011 Granada, Spain; fraso@ugr.es
* Correspondence: cmoral@ugr.es

**Abstract:** Successful school leadership in disadvantaged contexts is a topic of debate in the context of the Spanish education system. There needs to be clear identification of the concept of school success and what successful school leadership means. Thanks to the research carried out within the ISSPP project in Spain, we have been able to expand our knowledge on how success is defined in disadvantaged contexts: How have principals contributed to their school's success? How has success grown and been sustained? And how have successful principals overcome internal and external factors to build a strong professional identity that enables them to fight for social justice? The ISSPP research in Spain finally concludes that the leadership variable may be considered decisive in the learning outcomes of pupils. It is not only the context variable that would be decisive in academic results, but rather leadership capable of making the moral purpose of education a reality.

**Keywords:** successful leadership; school success; disadvantaged context

## 1. Introduction

In Spain it is assumed that leadership is a decisive variable for school success, and specifically, after the teaching staff variable, it is the second variable that contributes to a greater extent to the success of student learning [1]. It is also assumed that, in order to achieve school success, it is necessary to abandon merely organizational, administrative, and bureaucratic leadership and to implement a pedagogical and distributed leadership model: the pedagogical leadership model is defined as instructional leadership and is also associated with leadership that generates a common vision, a collective dynamic of working together, and a transformation of the attitudes, motivations, and behaviors of the teaching staff [2–4]. Furthermore, it is considered a distributed leadership model defined as leadership shared by the school community as a whole [5]. To this end, a Spanish Framework for Good Leadership has been established, framed within the parameters of pedagogical leadership as a guarantee of school success [6,7]. School success does not only refer to academic success, pass rates, repetition, and dropout rates; it also encompasses aspects of students' emotional and competence development [8]. Based on the recommendations of the DESECO (Competence Definition and Selection) project promoted by the OECD (Organization for Economic Co-operation and Development) [9] in place since the education law of 2006 [10] and ratified in successive subsequent laws, LOMCE (Organic Law for the Improvement of the Quality of Education) and LOMLOE (Organic Law on Education Improving the Organic Law on Education LOE of 2006) [11,12], the Spanish education system has adhered to the training of students in key competences. Currently, with the latest education law, LOMLOE [12], wherein the basis for establishing the curriculum at the national level is specified, the tendency to value competences has increased. As a result, the objectives, basic curricular knowledge, and the benchmarks

of success for primary and secondary education are articulated and subordinated to the acquisition of basic competences [13,14].

In disadvantaged contexts, school success is conceptualized from this same competency-based approach and from the same model of pedagogical and distributed leadership established by the Spanish Framework for Good Leadership [6,7], but with the added nuance that the leadership model applied in disadvantaged contexts must be leadership that encourages social justice [15–17]. Regarding this type of leadership, it should be kept in mind that in Spain, there are a series of determining factors that must be considered in order to identify what is understood by successful leadership for social justice.

Firstly, there are different definitions of social justice. According to [18] (p. 18), social justice is based on the concept of equal opportunities and can be assigned to three categories: (1) A meritocratic equality of opportunities where merit and personal effort are valued; (2) a universal or equal opportunity equality in which all students should be treated equally regardless of their talent or wealth; (3) compensatory equal opportunities in which affirmative action is used to compensate for inequalities and disadvantages. From this last approach, the principle of merit is rejected in favor of the principle of compensation, and it is recommended that the curriculum is equalized through key competencies. This approach to social justice follows the recommendation of Tedesco [19] in terms of eliminating the anchors associated with content and providing compasses—skills—to guide oneself in the knowledge society via basic competences training. For some, this implies establishing a problematic relationship between quality and equality [20] and contradicts the OECD's slogan Equity and Quality in Education [21].

Secondly, it should also be considered how different definitions of what school success means can be found. The study by Camarero et al. [17] shows how school principals in disadvantaged contexts may have different perceptions of school success. For some, it is achieving academic success, for others, it is making children happy and joyful; and for others, it is simply keeping them fed. For some authors, excellence in disadvantaged contexts is a term to be avoided as it is associated "with an elite school culture and represents the essence of conservative and neoliberal thinking in the field of education" [22] (p. 67). The slogans of excellence and quality in the 21st century [23] are, for some, a mere camouflage for a competitive, prevailing logic obsessed with raising the level of students' academic results [24].

Thirdly, there are several limitations to exercising pedagogical leadership [6]. For García [25], Spain is a country with very weak and unprofessionalized school management that prevents effective pedagogical leadership. The activity of the principal is reduced to merely technical/bureaucratic/administrative dimensions, with little initiative in terms of making improvements, for they are considered mere instruments for maintaining the system rather than agents for making improvements [26,27]. Spanish school principals do not manage to carry out the tasks of a pedagogical leader, thus generating nervousness and stress. They work long hours, have few rewards, their tasks are diverse and complex [28,29], and they have little authority in all matters relating to the professional development of teachers [30]. Principals also have a weak professional identity as they have a dual identity: they are both teachers and principals [31,32].

Finally, it should be noted that in disadvantaged contexts, it is assumed that the curriculum may be less demanding than in advantaged contexts. There is even talk of dual education for different groups of learners who are advantaged or disadvantaged [33]. This leads, in some cases, to a dual curriculum, which in turn becomes an instrument of exclusion [34]. To escape from this problem, the recommendations are to opt for a methodology based on Freire's critical pedagogy approach [35], together with the whole movement based on the acquisition of key competences [18].

## 2. Description of Reported School Case Studies and Review Methodology

This paper consists of a critical literature review [36] of different publications about the case studies carried out in Spain within the framework of the ISSPP project, which

aims to analyze the figures of successful principals in different contexts and countries. ISSPP has the following strands of research: Strand 1: Successful school principals. School principals of primary and secondary schools located in areas of social advantage and economic disadvantage who sustain success; Strand 2: Principals of schools which under-perform. Principals in visible and invisible under-performing schools; Strand 3: Principal identities. Principal professional identities; Strand 4: The evolution of successful leadership and how it is sustained over time; Strand 5: How successful leadership is implemented in a complex context. This approach seeks to analyze the complexity of leadership. The Spanish case studies have explored all these strands and lines of research. The critical literature review has been carried out on publications in Spanish and English language journals related to ISSPP Spanish case studies of successful and unsuccessful schools in disadvantaged and heterogeneous contexts, covering all the ISSPP lines and branches of research. Specifically, twelve case studies have been carried out by different research groups belonging to the universities of Huelva, Madrid, and Granada [37–43]. All case studies have been published in peer-reviewed high impact journals and have passed the required validity criteria for publication. Table 1 shows the different case studies carried out by the different Spanish research teams participating in the Spanish ISSPP, showing the characteristics of the different case studies carried out in different types of schools, the developing of different strands of research, and their corresponding publications.

**Table 1.** Characteristics of the case studies carried out, their settings, and the publications of the studies in journals with their corresponding ranking indexes (RIE—Revista de Investigación Educativa; SLM—School Leadership and Management; IJLE—International Journal of Leadership in Education; LPS—Leadership and Policy in School; NP—Next Publication).

| ISSPP Spanish Research Group | School Ownership | School Area | Level of Studies | Success Rate | ISSPP Strand | Journal Publication | Journal/Book Ranking |
|---|---|---|---|---|---|---|---|
| MADRID | Public | Urban | Primary | Successful | 1 | RIE | SJR Q2 |
| | Public | Urban | Primary | Successful | 1 | RIE | SJR Q2 |
| | Public | Urban | Secondary | Successful | 1 | RIE | SJR Q2 |
| | Public | Urban | Secondary | Successful | 1 | RIE | SJR Q2 |
| HUELVA | Public | Urban | Primary | Successful | 2 | SLM | SJR Q1 |
| GRANADA | Public | Rural | Secondary | Successful | 1/4 | SLM/JILE | SJR Q1/Q1 |
| | Public | Rural | Secondary | Successful | 1/4 | SLM/JILE | SJR Q1/Q1 |
| | Public | Rural | Secondary | Unsuccessful | 1/4 | SLM/JILE | SJR Q1/Q1 |
| | Public | Rural | Secondary | Unsuccessful | 1/4 | SLM/JILE | SJR Q1/Q1 |
| | Public | Rural | Secondary | Successful | 3 | LPS | SJR Q2 |
| | Public | Rural | Secondary | Unsuccessful | 3 | LPS | SJR Q2 |
| | Public | Rural | Secondary | Successful | 5 | Springer | Factor 0.4 |

### 2.1. Case Selection

The papers collected for the critical literature review of the Spanish ISSPP case studies were all those that followed the ISSPP research protocol for the collection and analysis of data from the various strands and lines of research offered by the ISSPP. The papers selected to carry out the critical literature review have used samples from public primary and secondary schools characterized by having heterogeneous and disadvantaged populations, and which are thus in need of social justice initiatives [35]. The schools in the case studies are compulsory from a mix of urban and rural areas. The schools were selected based on the criteria that the ISSPP uses to select successful/unsuccessful schools

located in disadvantaged contexts [44,45]. These criteria were as follows: (a) schools where parental academic and income levels are low are designated as serving disadvantaged communities; (b) student achievements through tests and examinations have been high/low with an ascending/descending progression compared to schools in similar circumstances; (c) schools that have received positive/negative school inspection reports regarding leadership. Many of these schools are located in disadvantaged economic and social contexts and host a diverse population, with numerous immigrant and Romani families with precarious employment or who are unemployed with very low educational levels. Other schools, although they do not have a Romani pupil population, are heterogeneous schools with a very diverse population.

### 2.2. Data Collection and Analysis Procedure

Once the Spanish ISSPP cases to be critically reviewed had been selected, the following analysis criteria were applied to carry out the critical review: (1) comparing the figure of successful and unsuccessful principals; (2) analyzing strategies for the sustainability of successful leadership; (3) searching for identity traits of successful principals; and (4) explaining successful leadership from the perspective of complexity theory.

The research carried out in the papers selected for the analysis of the critical review of Spanish ISSPP studies uses the methodological basis of case studies [46]. The case study protocols and questionnaires provided by the ISSPP have been used to conduct the Spanish case studies [37–43]. In particular, to carry out the Spanish ISSPP, the case studies used individual interviews with principals, teachers (3 to 6 teachers in each case study), and educational inspectors/counsellors; group interviews with teachers, students and parents; and document analysis, as per the core research tools, supplemented by questionnaires and observations of school life in some cases [44,47]. The interview questions focused on exploring the personal characteristics, dispositions, and leadership strategies used by the principals, always using as a reference both the ISSPP model of successful leadership [44,48] and complexity theory [47]. The process of analysis followed in the case studies of lines 1, 2, 4, and 5 has been carried out starting first of all from the ISSPP category system and, in addition, taking into account the codes that inductively emerge during the analysis of the data [49]. The analysis of the line 3 case study data has been carried out using the foundations of grounded theory provided by Strauss and Corbin [50]. The data analysis programs used by the different research teams (Madrid, Huelva, and Granada) have been as follows: -ATLAS.ti.7, NVIVOb10.1.3, MaxQda.

The analysis conducted in terms of contrasting case studies in this critical literature review will not be able to reach the kind of generality and implications of a quantitative study with a larger sample. However, following Flyvbjerg's reasoning [51], we believe that the case study is a research method that has sufficient validity to be considered a very valuable method for gaining knowledge about successful school leadership, and is a suitable method that holds up well when compared to other methods in social science research methodology.

## 3. Essential Findings in Relation to the ISSPP Case Studies Conducted in Spain

### 3.1. How Has Success Been Defined in These Case Studies?

As discussed in the first section, current approaches used to define school success in disadvantaged contexts in Spain are determined by different options for understanding what social justice means, by the objective being pursued in terms of excellence or non-excellence, and by the possibilities for carrying out a pedagogical leadership that promotes appropriate curriculum development (in a mix of centralized and decentralized curriculum).

The lesson learned from the analysis carried out in the ISSPP case studies in Spain is that successful principals in disadvantaged contexts conceive success in a way directed toward excellence from equity [39–42]. They seek excellence from equity "almost like an obsession" [40] (p. 570). For underperforming principals, the objective may be "to be on time, to come to class clean, to talk with respect... the objective is not for the kids to

graduate—no, we've got to start with the basics" [41] (p. 42). In underperforming schools, a double curriculum can also be observed [34] since it is common to find comments such as: "some students are required, but not all equally" [42] (p. 123). There is academic interest regarding some students, but not with all.

In unsuccessful schools, it is common to find comments that identify context as the reason and cause that triggers differentiated treatment in terms of students. To avoid contextual differences, they establish a series of procedures that, in turn, serve to justify their unequal action, opting to make adaptations and establish different levels of demand according to the possibilities of each student, since otherwise, as they say, "it would be impossible" [42] (p. 123), to work with such a diverse and unequal student body, although this is the basis to begin to establish social differences and deepen social class inequality [52–54].

The successful principals analyzed do not differentiate the treatment of students according to their context, nor do they choose one of the three alternatives proposed by Bolivar [18] as the only possible alternative. For successful principals, all the options would be present in some way, as they value merit and individual effort of students, they do not differentiate curricula according to talent, and although they propose an education based on competencies, as indicated by the requirements of current educational administration, they do not abandon the importance of content as central to their planning and curriculum design.

The motto is 'excellence and quality through equity' [21,23], terms that seem to conflict, as they can be seen as opposites [55], but which successful principals manage to fit together suitably. The explanation that one of the successful principals interviewed considers to be the reason for being able to maintain the slogan 'excellence and quality through equity' is that in the school, there are agenda and people [43]. Firstly, for a school to be successful, there must be an agenda that functions as a brain, which provides meaning, direction, and vision and sets the course for where the school is heading. This agenda is elaborated and sustained by the principal and the management team. It also needs enthusiastic and passionate teachers who share the responsibility of providing quality education for all pupils equally [56] to make the agenda a reality [43].

Therefore, we have learned from the ISSPP that it is not the context that determines success, but rather it is the agenda and the people that make success possible: those who are able to change the prejudices and barriers that prevent the true integration of diverse and heterogeneous students without lowering the quality of teaching. This is the key to the transformation of schools and the beginning of the transformation of society and social justice.

### 3.2. How Have Principals Contributed to Their Schools' Success?

First, in order to talk about how principals in disadvantaged contexts contribute to the success of their schools, and who have an impact beyond the pupils' performance [57,58], one should start by talking about the characteristic traits, values, or qualities that identify them. These traits are attention and care for the most disadvantaged, respect for multiculturalism, responsibility, and moral purpose [15,16,35]. The research carried out at ISSPP has shown that both successful and underperforming principals share the same value system associated with a high sense of responsibility, commitment to the community, and social justice. They are also empathetic, approachable, and open to dialogue. However, in addition, successful principals also hold high levels of academic optimism, intrinsic motivation, resilience, and passion for their work [37,38,40–42], maintaining, in the end, an attitude of strength and control [40–42]. This control is not associated with a rigid control that coerces freedom but rather a control that is engagement.

Successful principals keep an optimistic vision in terms of improvement and communicate a coherent organizational vision and clear goals. They convey this desire for change while maintaining a firm conviction that making progress towards social justice does not take them to a dead-end street but to a difficult, albeit promising, path, which

requires them to have an attitude of vocation and service to the educational community and society [39–42].

In relation to the strategies used in the Spanish context, it is recommended to bet on pedagogical leadership with distributed and transformational traits: distributed leadership is identified with shared leadership [5]; and transformational leadership is identified as a leadership aimed at transforming current ideals, motivations, and values of school staff, increasing their expectations in terms of improvement and their commitment to organizational goals [59,60], and in disadvantaged contexts struggling for social justice, the application of this model is highly recommended [15,16]. The dimensions that identify the leadership model established as a benchmark by the Framework for Good Leadership in Spain [6,7] are closely related to the practices of successful leadership highlighted by the ISSPP analytical framework: establish direction, build relationships, design the organization to support desired practices, manage and direct the teaching–learning program, and ensure accountability [47,48,61,62].

However, what we have learned since the time of the Spanish ISSPP research is that the relationships that are established between these successful practices are not linear, they are complex, and a type of leadership capable of understanding this complex process is required [43]. The leadership model recommended by the Spanish Framework for Good Leadership [6,7] is conceived as a model for direct application of pedagogical leadership. Research carried out by ISSPP shows how the complexity of school leadership requires the application of a diversity of types of leadership, such as leadership attentive to community demands [63], an humanitarian, emotional, and nuanced leadership [47,64], and adaptive leadership [65–68], combined with instructional [69], collaborative [70,71], and innovative leadership, through the implementation of action research processes [68,72]. Therefore, to implement effective leadership, not only the application of one type of pedagogical leadership is required, but also the use of other types of leadership working in a unitary and joint way, complementing each other, to generate a process and a dynamic that generates school success.

### 3.3. How Has Success Been Grown and Been Sustained over at Least a Three-Year Consecutive Period?

In Spain, there are no studies or precise indications on how to maintain success in schools. By contrast, the ISSPP has conducted in-depth research on sustainability in leadership, carrying out numerous investigations in order to check the characteristics and strategies of successful principals maintained over time [73–78]. In particular, the ISSPP has studied how to sustain the 'turnaround' in schools serving challenging communities [79–83].

Day et al. [84] point out that successful leadership aimed to improve learning results goes through a series of phases that progress from generating an environment conducive to learning and distributing leadership to personalising and enriching the curriculum. This improvement process is not linear, but it happens in layers, called 'multilayered'. Layers are supported among themselves, and some are necessary for others to be generated [62,74]. In addition, the research shows that the sustainability of a successful leadership requires the active implication of all school members. A heroic leader cannot sustain and maintain the improvement on his/her own. The results of the ISSPP speak of a post-heroic leader who distributes leadership effectively, builds a collaborative professional culture, creates a learning community, generates a collective responsibility, and guides the organization through the establishment of clear objectives firmly set [85].

The results obtained from ISSPP research carried out in Spain confirm these findings, delving deeper into this post-heroic leader figure that sustains success. As we have considered in the first section of results, the successful principal is fully aware that the basis of success for their school concerns the existence of a united group of people who support and sustain the school's agenda. However, at the same time, there must be a system of accountability connected to research–action processes that promote and sustain constant improvement to make the curriculum more attractive and personalized to the diversity of

students in the school. Improving and innovating the curriculum also involves responding to external and changing indications from the educational administration; for this reason, the principal always studies new educational regulations in order to adapt them without abruptness to the progress and development of the school [43].

The key is to maintain and sustain the school project. The school project must be cared for and nurtured as something alive that requires much attention [42], especially taking care of, and giving encouragement to, the teachers who are the ones who put the school agenda into practice. It is not only about maintaining the system but improving it, fostering a culture of 'it can be done' [86], not resigning themselves to accept to a certain degree that 'this is how things are', assuming a position that they refer to as a realistic approach, where they accept that the curriculum is fundamentally aimed at achieving social inclusion [42]. The fact is that, as Capper and Young [52] point out, this type of leadership for inclusion is merely a label since some leadership practices, although they may seem to promote social inclusion at first glance, are not truly inclusive. This is because, ironically, they may exclude some pupils without this being questioned from the perspective of social justice [52]. Injustice not only arises from unfair treatment that pupils might receive but also when it is assumed that as far as pupils from disadvantaged backgrounds are concerned, the curriculum should have a simpler focus and a less rich program [54]. Therefore, what we have learned from ISSPP is that we must overcome this realistic approach and move towards a realistic utopian approach that looks to the future with the hope of changing things and is inserted in proposals for research–action and constant improvement [42,43].

### 3.4. Do the Ways That Principals Contribute to Success Vary in Different National, Workplace, and Principal Capability Contexts? If So, How?

In Spain, the principal is conceived as the public authority of the school so that he/she can autonomously elaborate the rules for its organization and functioning, although some question his/her authority [30]. In any case, the principal is not conceived from an authoritarian and power-driven perspective to manage the school. The figure of a charismatic leader/manager who controls teaching tasks has been widely criticized in the Spanish context. The principal is conceived as more of a facilitating leader with transformational, collaborative, democratic, and distributed leadership traits [5,59]. The ISSPP also moves away from the figure of a charismatic and dogmatic leader. However, based on the research carried out in ISSPP in the Spanish context, several other traits have been found in relation to the way of understanding the role of the authority, voice, and power of the principal, as well as the forms of distribution of leadership:

1. First, ISSPP research in Spain has found, like Day [74], that successful principals are not charismatic or heroic in the traditional sense, but they have a very resolute sense of their moral purpose and personal characteristics that have become a benchmark. Principals are true post-heroes who generate communication relationships based on trust and collaboration that allow the development of a distributed leadership. However, we have found from our Spanish ISSPP findings that teachers seek or desire a principal figure with traits of direction and authority, although not authority in a negative sense, but rather associated with a high degree of knowledge and control of what happens in the school. They like principals with traits in the direction of a convincing or assertive character, because without these traits, they are somehow confused. This trait is associated with the capacity of the principal to build a vision toward clearly identified goals with a commitment to improvement [87]. Without this unifying vision, schools are destined to be systems with a very slight connection, characterized by a number of individuals located in a common area with almost no interaction and without a common purpose [88]. One teacher describes the situation that occurs when a principal with these traits is removed from the school: "He was like an orchestra conductor. . . With this new principal, everyone in their own way, and everyone is as if playing an 'intuitive music'" [42] (pp. 119–120).

2. Successful principals shared power with the teaching staff. Their authority as leaders did not consist of creating the feeling that they were at the top imposing their

criteria, but rather of fostering a feeling of sharing power with the teaching staff. Power is shared in collaborative decision-making, but also by delegating power directly, seeking that teachers also become leaders, keeping them informed, motivating them, and engaging them in the school's educational project. Therefore, it could be called shared leadership [71,80], as successful principals set up an institutional plan for systematic and collaborative decision-making [82,89].

3. Successful principals empower the figure of the teacher leader and favour their prominence because they recognise that the most powerful force in a school is the teachers, as they are the heart that drives the energy to get the school going [43,47,56]. Nevertheless, even though the successful principal follows a model of democratic and collaborative leadership, the principal delegates functions and distributes leadership in a selective manner. In public schools in Spain, principals cannot select their teaching team, but the successful principal somehow selects the members who will be middle leaders, such as members of the management team, department managers, and tutors, to whom the principal will delegate functions to manage the school structure. The principal selects teachers based on their professional competences and their motivation towards innovation. To make this selection, the successful principal carefully observes the skills, competences, and interests of the school's teaching staff. It is like making an X-ray of each teacher in order to know where he/she would be most suitable for the development of the school agenda [43].

*3.5. Have Internal or External Factors 'Caused', 'Shaped', or 'Influenced' the Ways That Principals Contributed to the Success of Their Schools?*

The agency and identity of the school principal are determined in the Spanish context using a series of external factors. In the first place, educational laws and changes in the regulations enacted by the educational administration [10–14] can be highlighted as determining factors in the performance of these professionals. However, there are also other factors that determine their performance and shape their professional identity. According to the Spanish Principals' Federations, there is a lack of objectivity in the system for electing school principals, as it is corporative and reinforces aspects such as seniority in the post over specific training [6]. These aspects determine the figure and performance of school principals, as they are deprived of sufficient training to carry out their tasks with confidence and competence, weakening their professional identity. Together with the failures highlighted by the Spanish Principals' Federations [6], we find the contribution of Bolívar and Ritacco [31] and Ritacco and Bolívar [32], who consider that school leadership in Spain is not a motivating or attractive task, adequately remunerated and with real possibilities for promotion. Furthermore, these authors point out that Spanish school principals have difficulties in acting according to their own identity, as they have a dual identity, with the paradox of being both a teacher and a principal at the same time [31,32,90]. For all these reasons, school principals present an unstable, fragmented, complex, and heterogeneous identity [31,32,90]. This lack of identity has negative repercussions on their professional practice and on the development of pedagogical leadership for improvement, leading principals to limit themselves to administrative management tasks instead of pedagogical leadership tasks.

However, in the research on the identity of successful Spanish principals, we have found that successful principals do not have an unstable or insecure identity. Rather, we could say that successful principals have a well-defined, stable, and secure identity, developing an agency with individual and social traits that define the performance of a successful leader according to ISSPP [39,91,92]. In contrast, the identity of underperforming principals does not reflect a clear connection with the role of a leader as described by the ISSPP, and their identity is superficial and associated with a merely bureaucratic role. It is a contingent and unstable identity conditioned by contextual variables such as the socioeconomic level of the families attending the school, the lack of resources, educational regulations, political variables, etc. [39]. The professional identity of successful

principals is not just situational or socially determined, conditioned by the uncertain contextual or educational conditions that current Spanish society experiences. Rather, it has higher purpose ambitions and they are more certain in the knowledge that they can make a difference to the lives of students. It is not easily distracted from this purpose by contextual matters like changing social and educational circumstances, applying a style of leadership that has, to some extent, to do with entrepreneurial leadership [93]. The identity of the successful versus underperforming principals seems to be different, with successful principals more hopeful and optimistic and underperforming principals more trapped by the many contextual and social variables. The successful principal builds his professional identity from a realistic utopian vision focused on improving students' learning results. On the contrary, the underperforming principal uses a merely realistic view to interpret the school context's reality and make sense of the principalship [39].

## 4. Conclusions

The research carried out in Spain within the ISSPP project provides a clearer appreciation of the problems associated with the quest for successful leadership in disadvantaged contexts. Not all Spanish principals analysed from the ISSPP perspective share the same idea of success and, therefore, do not all solve the problems highlighted above and act in the same way. Depending on the degree to which they can be qualified as successful or unsuccessful, they show a series of characteristic traits, strategies, and identity notes that differentiate them. Therefore, based on ISSPP, we can have a better delimitation and conceptualisation of successful leadership, as well as appreciate the nuances of what successful leadership means in disadvantaged Spanish contexts. These results may help to solve the problems that determine current Spanish education in disadvantaged contexts. Although carried out with a reduced sample, our research shows that the leadership variable may be considered decisive in the learning outcomes of the pupils. It is not the context variable that would be decisive in pupil success, but rather a leadership capable of involving teachers; an open and optimistic leadership with hopes and expectations, making the moral purpose of education a reality. In short, the final conclusion of the Spanish ISSPP adds to the ISSPP project's earlier stated assertion that effective leadership lies in "the commitment to provide a high-quality education for all students regardless of background, rejecting... as a resolute failure, the acceptance of context as a determinant of academic and social success" [62] (p. 242). The Spanish ISSPP will continue to advance in this direction in order to decode the complex systems that facilitate or impede the development of quality, success-driven education for all children equally.

**Author Contributions:** Conceptualization, C.M.-S.; investigation, C.M.-S. and F.R.-S.; All authors have read and agreed to the published version of the manuscript.

**Funding:** This research received no external funding.

**Institutional Review Board Statement:** Not applicable.

**Informed Consent Statement:** Not applicable.

**Data Availability Statement:** Not applicable.

**Conflicts of Interest:** The authors declare no conflict of interest.

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
