# Peer review of "The Meaning of Successful School Leadership in Disadvantaged Contexts in Spain: Approach from the International Successful School Principalship Project (ISSPP)"

_education, doi:10.3390/educsci13101007_

Round 1
Reviewer 1 Report
The paper is well structured and reads well. However, in the first section, lines 31-34 there are a lot of acronyms that should be spelled out the first time they are mentioned. Further, the term "transrformational leadership" is used three or four times in the paper, and I would like to see a short definition of the term, as many researchers use the term with different meanings. This would be very helpful for the international readership of the journal. Finally, in Section 3.5, where the author(s) discuss Internal and External Factors, I think that the paper would benefit if they read the following work, which deals effectively with this issue, and calls it "Edupreneurial Leadership":
Pashiardis, P. & Brauckmann S. (2019). New Public Management in Education: A Call for the Edupreneurial Leader?, Leadership and Policy in Schools, 18 (3), 485-499. DOI: 10.1080/15700763.2018.1475575.
I think that, in general, the paper needs some minor editing for English Language, especially the title so that it makes the paper more appealing. I would refrain from giving examples myself. I am sure that the author(s) can think of something more original.
Author Response
RESPONSE TO THE FIRST REVIEW
Thank you very much for the revision of the article which has certainly improved it. The response to your comments in the following:
- The abbreviations of the acronyms have been indicated.
- It has been explained that means transformational leadership.
-The entrepreneurial leadership style and the reference to Pashiardis, P. & Brauckmann have been included.
-The title, we are sorry, has not been changed. We recognise that the title is not at all original, but it says exactly what the article is about and this is what we mean.
-The paper will be sent for revision of the edited text.
Reviewer 2 Report
Dear Authors,
I found this paper very interesting, and my assessment is publication after some major revisions.
The concepts of pedagogical leadership and distributed leadership should be elaborated and clarified since they are both broad sensitizing concepts and widely covered from different angles in the research literature. Moreover, the case study approach, as applied in this study, should be clarified. For example, what is a "multiple perspective case study methodology"? Thirdly, I know something about the ISSPP project, but all readers do necessarily not share this knowledge. So please add a note on this unique project. It might be useful to have a look at a Swedish longitudinal case study published by Jarl, Blossing, and Andersson- one that illuminates organizational characteristics found in respectively high-performing and low-performing schools.
As you can see, English is not my mother language, yet I hope my points are understandable.
Sincerely
The Reviewer
Jarl, M., Andersson, K., & Blossing, U. (2017). Success or failure? Presenting a case selection strategy for studies of school improvement. Education Inquiry, 8(1), 17-32. doi:10.1080/20004508.2016.1275177
Jarl, M., Andersson, K., & Blossing, U. (2021). Organizational characteristics of successful and failing schools: a theoretical framework for explaining variation in student achievement. School Effectiveness and School Improvement. doi:10.1080/09243453.2021.1903941
Author Response
RESPONSE TO THE SECOND REVIEW
Thank you very much for the revision of the article which has certainly improved it. The response to your comments in the following
- The concepts of pedagogical leadership and distributed leadership have been clarified.
- The concept of multiperspective case study has been removed, it has been replaced by a basic reference to case study.
- The methodological part has been clarified to better explain the methodological basis of the ISSPP project.
- The way of selecting the sample of successful and unsuccessful schools based on the criteria of the ISSPP project has been indicated.
- The paper will be sent for revision of the edited text.
Reviewer 3 Report
Dear Authors,
I enjoyed reading your manuscript ‘The meaning of successful school leadership at disadvantaged context in Spain: Approach from the International Successful School Principalship Project (I.S.S.P.P.). You provided a thorough discussion of previous ISSPP research related to leadership in Spain and elsewhere as well as limitations of the literature that ISSPP studies have filled. I appreciated the description of the policy context and the role of leadership in Spanish schools. I also appreciated the description of social justice literature as it relates to leadership in schools.
I have the following suggestions to improve the manuscript for publication in the special issue on ISSPP in Education Sciences.
First, it would help an international reader to have additional description of the context for education, curriculum, and pedagogy in Spain. You advocate for pedagogical leadership, but I am not clear if this means that principals need to mediate between a national or centralized curriculum and the needs of all students, including particularly those from disadvantaged circumstances.
Closely related, the article would benefit from a few sentences to describe the student demographics in Spain. Have demographics changed over time? How, if at all, have demographic shifts contributed to student disadvantages? What are these disadvantages and do they vary in terms of school location (e.g., rural, suburban)?
Second, the article needs more detail on the methods used to re-analyze findings from ISSPP studies in Spain. The authors provide good detail on the ISSPP methods used to select and develop case studies, but it is not clear how the cases were re-analyzed to present the synthesis of the findings.
Third, I am interested to learn more about next steps for ISSPP in Spain. You write about layers of influence and increasing complexity as these relate to schools and leadership. What are your plans to gather additional cases in relation to the new theoretical framing and mixed methods research design?
Finally, I suggest getting an editor to do an English edit. There are minor errors in several places of the article.
I suggest getting an editor to do an English edit. There are minor errors in several places of the article.
Author Response
REPLY TO THE THIRD REVIEWER
Thank you very much for the revision of the article which has certainly improved it. The response to your comments in the following:
- The methodological section has been revised and completed for better understanding.
- An indication has been made that the Spanish curriculum is a national curriculum. The demographic issue could not be considered as we believe that it is beyond the scope of the article.
- A clarification of the methodological section has been made.
- A brief commentary on the future of Spanish ISSPP research has been indicated at the end of the article.
- The paper will be sent for revision of the edited text
Reviewer 4 Report
The topic of this paper is relevant and likely to provide important knowledge to the field because we need more large scale studies from countries other than the U.S. to add to the knowledge base on educational leadership. The aspects of the discussion that reference leader identity are strong and provide evidence to support the ongoing professionalization of the principalship.
The findings of this study, therefore, suggest this article could make an significant contribution, but its current state is more appropriate for a professional journal where the author's expertise is assumed than an empirical journal where the researcher must convince the reader that the author's conclusions are warranted. I offer the following suggestions to remedy this and add more clarity to the reader:
-
Provide context about the International Successful School Principal project and the law that was referenced.
-
There needs to be much more transparency about the analysis and what the data said so the reader understands how the authors’ conclusions were drawn.
-
The literature on collective efficacy, teacher sense-making, and deficit orientations are salient to the study and could further contextualize the results. I also see connections with the 2021 Grissom et al. 20-year review of research on how principals affect schools.
-
The methods are murky. We need to know more about the case studies: What are the cases? How many individuals were involved? Where did the data come from? What methods were used to analyze the data? How were disadvantaged social contexts and diverse populations defined in selecting the samples? How were “effective” principals defined?
Author Response
RESPONSE TO FOURTH REVIEWER
Thank you very much for the revision of the article which has certainly improved it. The response to your comments in the following:
- The basis of the ISSPP project has been explained more clearly and the methodological basis of the ISPP project and the article has also been explained in more depth.
- The methodological section has been revised and completed for better understanding.
- The reference to the Grissom et al study of 2021 has been added.
Round 2
Reviewer 2 Report
Dear Authors,
I have enjoyed reading the revised version of the manuscript, and am indeed looking forward to reading the article in technically published form in this journal.
Sincerely, the reviewer
Author Response
Thank you for your review.
Reviewer 4 Report
The authors addressed many of my prior concerns and have provided more information about where they got their data. The study has the potential to offer valuable insights into the field but it does not currently meet some key criteria of empirical studies. I have the following suggestions for the authors to consider:
1. Now that it is clear that this is a literature review, the reader needs to know more about the quality of the literature from which conclusions are being drawn. Some ways to do that could be:
- A table showing the 12 studies and providing information about their sites and samples.
- Related: Confirmation that the 12 studies are unique contexts and multiple papers are not outlining the same school/schools.
- A more detailed explanation of how the search to locate the papers was conducted with inclusion/exclusion criteria. Were these studies published in peer-reviewed journals? How can we be assured that the researchers followed the ISSPP protocols?
2. The exact methods the authors used to locate patterns in the data are still under explained. Was this inductively coded? Who did the coding? What steps did the authors take to mitigate their own biases?
3. If the authors are to make claims about "successful" and "unsuccessful" principals, it is important to make explicit whether the definition of successful leadership is having positive school inspection reports (or something else.) If this is the case, the international reader needs some context about who does the inspections, what the potential outcomes are of the inspections, and which outcomes constitute "success."
4. The reader should be made aware of the study's limitations. In particular, these findings seem to be drawn mainly from perception data that does not support causal inferences. These also appear to be small case studies so the authors should refrain from making broad claims about leadership.
5. The findings should be more contextualized by the wider literature. The Grissom study that was recommended to the authors was briefly referenced, but the authors missed the opportunity to show that this strong review of 20 years of leadership literature supported some of their findings (e.g., building a productive school climate and the way leader support equity.) There are other broad studies about educational leadership that would have strengthened the authors' claims (e.g., Louis et al.'s Learning from Leadership project.)
There is some terminology that is overly definitive for an empirical paper. There are also some terms that are used in ways that are confusing to this native English speaker. Consider revising:
- "Complexity Theory" is a specific theory that was mentioned as a topic of the paper, but was not explored in the study and should probably be removed.
- "To avoid contextual differences" on p. 4 does not make sense with what follows.
- "His motto" on p. 4 assumes that all principals are male.
- "Perfectly" on p. 4 is too strong a term.
- Use of the "project" terminology is confusing. A better term might be "agenda."
- p. 5 "Closeness and listening" are not forms of control. "Engagement" might be a reasonable substitute.
"All the time" is not possible.
The principals cannot be both "not heroic" and "true heroes." Also, labeling the principals as "heroes" violates the unbiased stance of the researcher.
- Instead of "control" or "firmness" the word "direction" would be a better fit for what the authors described in this section.
- The constant reminder of the Spanish context became distracting.
- The "realist" verses "realistic utopian" point was unclear and repeated twice. The authors seem to view "realist" in negative terms for reasons not clear to the reader. Many readers may view "realistic" and "utopian" as an oxymoron because Utopias are impossible to achieve and see no meaningful differences between "realist" and "realistic".
Author Response
Second response to the fourth reviewer
- A table of the case studies carried out has been introduced giving more information about the schools and the journals in which the case studies have been published. The information in the table answers several questions about the timeliness of the analysis carried out, as all publications collected for the critical review have been conducted with high impact journals that require peer review and have a high credibility.
- A clarification has been made regarding the type of analysis carried out following an inductive coding and grounded theory analysis process.
- In relation to the questions for the identification of successful and unsuccessful schools, the selection criteria used by Notman and Henry (2011) and Day (2013) in the previous review of the work were indicated, but I am unable to provide information in relation to the inspection reports. The inspectors' response was a reply in which they only gave the names of the schools they considered most successful and least successful, but they did not give us a written report in relation to the schools, so we are unable to respond to this question.
- In relation to the limitations of the study to support causal inferences, we have made a digression in relation to the validity of case studies which do not pretend to reach generalisations like quantitative studies, but are very valid in social science research.
- We have included reference to the research carried out in the Learning from Leadership Project.
Response to editing coments
All comments on the editing of the paper have been resolved and modified.
I hope everything is correct, if this is not the case please do not hesitate to contact me.